# Loyalty to World Heritage Cities: A Case Study of the Historic Ibn Ṭūlūn Mosque in Cairo (Egypt)

Ibrahim Mohamed Ibrahim Elassal [1] , Ricardo David Hernandez-Rojas [2,*] and Luz Arelis Moreno Quispe [3]

1   Sinai High Institute for Tourism and Hotels, Cairo 32923, Egypt; ibrahimelassal2@gmail.com
2   Area of Financial Economics and Accounting, University of Cordoba, 14004 Cordoba, Spain
3   Institute of Research for the Sustainable Development of Tourism, University of Frontera, Sullana 20103, Peru; lmoreno@unf.edu.pe
*   Correspondence: et2heror@uco.es

**Abstract:** The city of Cairo is famous worldwide for its distinct heritage sites. Its uniqueness, its call effect, and its history are significant. So, it is essential to improve its management of tourism. The main objective of this research is to analyze the satisfaction of visitors to historic mosques in the context of heritage tourism. The study focuses on tourist loyalty from several perspectives: expectations, satisfaction with the city itself and satisfaction with the mosque. The fieldwork was carried out in Cairo (Egypt), specifically in the Ibn Ṭūlūn Mosque. It is more than 2000 years old and is considered the oldest mosque in Cairo and one of the most amazing mosques in the world. There are few studies on mosques from a tourism perspective, which makes this study innovative and a substantial contribution to the literature. The methodology used was based on partial least squares (PLS) structural equation modeling. A structured questionnaire was used to achieve the proposed objective. The results obtained confirmed that satisfaction during their visit to the Ibn Ṭūlūn Mosque positively influenced tourists' loyalty, and therefore they recommended revisiting it.

**Keywords:** loyalty; heritage tourism; management; mosque





## 1. Introduction

The Greek historian Herodotus was quoted in the 5th century BC as saying, "Egypt is the gift of the Nile". His statement conveys the vital role the Nile played in establishing and sustaining all civilizations that have passed through Egyptian land. Egypt is considered one of the centers of world heritage [1]. Various fascinating cultures have dwelt, survived, and thrived on its land. Many cultures and civilizations are preserved in monuments, inscriptions, and writings in Egypt. Because of Egypt's many inhabitants and cultures over thousands of years, there is an enormous quantity of different architecture there. It contains an invaluable variety of cultural markers from all over the world, including Pharaonic, Greek Ptolemaic, Roman, Persian, Byzantine, Coptic and Islamic architecture. All of these cultures are fascinating and mysterious [2].

Throughout the thousands of years of Egyptian history, there has been significant expansion in terms of culture, science, architecture and art. Egypt was uniquely situated to become one of the oldest empires. It started with the unification of Upper and Lower Egypt in 3150 BC, which continued until 332 BC, when Egypt was ruled by the Greek leader Alexander the Great. Ptolemies ruled after Alexander, and then the Roman Empire ruled from 30 BC until the Muslim conquest of Egypt in 641 AD. Cairo was occupied by various Muslim dynasties that played a prominent role during Fatimids, Ayyubids, Mamluks, and Ottomans [3].

The city is known as "City of the thousand minarets", due to its unique peculiar skyline. Cairo has been classified by the United Nations Educational, Scientific and cultural organization, "UNESCO", as one of the "Cities of Human Heritage" since 1979 [4]. Many monuments still stand, pointing to the city's glory, especially in the area between the

mosque of Amr in the South and Bab al Nasr and Bab al-Futuh to the North. Outside this area, very few medieval buildings of interest have survived. Egypt's capital, Cairo, has had its name since the Fatimid period. Before the Fatimid period, Cairo was known by many other names. First, in 641 AD, it was known as Al-Fusṭāṭ. By 750 AD the Abbasid foundation had taken over, and the city was called Al'Askar. In 868 AD, Ahmed ibn Tulun found Al'Askar, and the Tulunid establishment fought for its independence from the Abbasid caliph. Therefore, the city was renamed Al-Qata'i in 869 AD [5]. In the city of Cairo is the mosque of Aḥmad ibn Ṭūlūn [6].

There are many archaeological and heritage sites in Cairo which date to different prehistorical and historical periods, in addition to numerous museums like the Egyptian Museum, overlooking Cairo's Tahrir Square, the Grand Egyptian Museum (GEM) in Giza, the Coptic Museum and the Islamic art Museum, which participated in flourishing cultural heritage tourism [7]. The mosque of Aḥmad ibn Ṭūlūn is the oldest intact religious building, dating to the Islamic era.

Consequently, there are different academic studies on the heritage in Cairo from historical, geographical, and architectural perspectives, but not on the historic mosques from the tourism perspective. At a general level, the studies carried out on cultural tourism have focused on cities that are world heritage sites; due to their high potential, they combine culture and material heritage, where the variables of satisfaction, valuation and loyalty are studied. Specifically, from this perspective, there are numerous studies that conclude that tourist activity linked to cultural heritage offers a unique tourist experience expectation [8,9]. In this sense, it is key that managers focus on the market for tourism assets that contribute to loyalty, satisfaction or expected value. This research is important because it presents these concepts from the perspective of management, since they help to improve tourism [10]. Likewise, the increase in pollution in cities is due to the lack of adequate mitigation options in the tourism sector. Improving mitigation options is a crucial challenge for cities wishing to increase tourism and enhance the quality of their tourism [11].

As a model used to achieve the proposed objective, the American Customer Satisfaction Index (ACSI) was used. It is a contrasted model used in the literature [12]. The model shows convincing precedents that have been used in the tourism sector [13,14] and can be successfully adapted to various areas in which the satisfaction and loyalty of the users of a good or service are studied [15].

In addition to contributing to knowledge about the loyalty of a destination with a strong material heritage component through heritage studies, this article contributes to the existing literature in two ways. First, it conducts a literature review, and second, it provides insights into the management of world-recognized heritage sites. The establishment of these objectives allowed the researchers to refine the available information on the complexity and diversity of heritage types in the city of Cairo.

## 2. Literature Review

### 2.1. Heritage in Cairo

Cultural heritage tourism in Cairo is considered one of the most researched topics in tourism literature. Cairo's heritage is known for its richness and diversity; besides its numerous archaeological and historical sites, different regions are distinguished for their intangible heritage. There are strong motivations for travelers to visit Cairo after the enormous growth of heritage tourism worldwide [16].

Although Cairo has such great significance as a cultural resource tourist attraction, many threats exist, including conserving and managing sites, as well as criminal activity like looting and illicit digging, which increased after the political troubles in the last decade [17]. The government is intensively promoting new solutions to archaeological discoveries and findings to international media in order to attract more tourists to the country [18].

Several studies present different views on the cultural heritage in the Egyptian capital, as well as the products of Islamic civilization. One of them is the mosque of Aḥmad

ibn Ṭūlūn [6,19–21]. Finally, concepts inherent to heritage and tourism such as sustainability, renewable electricity generation and environmental concerns must be taken into account [22].

### 2.2. Expectations in World Heritage Cities

Studies about the interactions that can take place between a tourist and a location are found in the literature. Studies have revealed that visitors and tourists value and assess several facets of their interactions with a location or company, and various criteria are taken into account in the resulting relationship [23]. There are numerous aspects of the overall management that the visitor or tourists analyzes. This is significant because no single factor may decide the value that a visitor places on a location [24]. Understanding how the service providers or products are marketed is crucial because it helps build and sustain client relationships, which can then influence outcomes like higher sales and increased word-of-mouth advertising [24]. According to the literature on the theory of commitment or loyalty, the expected quality [25], perceived quality [26] and satisfaction [27] are the most crucial elements in the relationship between a Muslim tourist and a location [28].

With respect to expectations when visiting a world heritage city, the literature agrees on the high expectations for tourist satisfaction [29]; this makes it harder for these types of cities to stand out, since the higher the expectations, the more difficult it is to surprise those who visit [30]. Consequently, this research helps visitors to consider other heritage sites such as Cairo and the historic Mosque of Ibn Ṭūlūn.

### 2.3. Satisfaction and Loyalty When Visiting World Heritage Cities

The visitor's perception of value in the context of tourism is regarded as the most significant predictor of their willingness to return to their location of choice. Numerous studies [31,32], obtained a realistic idea of tourist behavior by means of this variable after making decisions. The results obtained may be useful for public institutions and tourism providers when it comes to assessing the general usefulness of the visit based on the perceptions received.

Most published studies on heritage reveal a direct, significant link between perceived quality and repeat visitors [33,34]. Tourists are more likely to recommend or return to a place if there is a favorable correlation between perceived quality and their level of enjoyment [35]. According to other field investigations, the intention to suggest that others visit is positively impacted by the favorable perceived quality [26].

General satisfaction and visitor expectations are correlated with the economic and social development of the World Heritage sites. They consequently relate to the impact of an international call that is brought about by being included on this list [36]. Given the various strands that support its inclusion on the UNESCO list as a World Heritage, the research of destination satisfaction is appropriate (uniqueness, exceptionality, distinctive feature, universality, authenticity, among others). This makes it an important tourist destination [37]. Studies linking tourist satisfaction have shown that visitors' pleasure when visiting the place is influenced by their satisfaction with various aspects [38]. One of the most crucial factors for returning and, specifically, the visitor's loyalty, is the comfort level displayed by the tourist toward the destination and the monument visited [39].

Other studies demonstrate that customer loyalty and its impact on happiness, from the standpoint of the city and its heritage materials, provide a motivation for the visitor to suggest the location to others and to return to it [40–44]. Tourists who are happy with their experience are more likely to return to the same location and recommend it to their friends, family and acquaintances. This a component of commercialization [45]. Being loyal to a world heritage city is essential because it increases tourism seasonality. In light of this, various authors [46–48] agree that destination management needs to be improved. This affects both overall tourist satisfaction and tourist loyalty, i.e., it ensures that tourists return and suggest the place to other potential visitors. From a cultural point of view, a report stated [49] that the mosque has had a significant loss in terms of overall attendance

compared to other world heritage sites. They conclude that new ways of structuring visits from a local perspective are needed, given that most visitors stay for a short period of time.

Satisfaction is directly correlated with loyalty to a location [50]. As a result, the motivation for studying satisfaction is consistent with how this variable relates to other ideas, including loyalty. Loyalty is later defined as the point at which a visitor bonds with the destination and expresses their desire to return and recommend it to others [51].

## 3. Materials and Methods

Recommending the destination after the visit, returning to the destination and boasting about the tourist visit are all important for destinations. For destinations with a strong heritage component, these factors are more significant since they enter into competition among other similar destinations. With this objective, the expectations are analyzed, both in relation to the destination of Cairo (ExpectCairo) and to the historical Mosque (ExpectMosq), along with the destination satisfaction (SatMosq) and the historic mosque itself (SatisMosq) individually. The model proposed for this study used five variables to measure loyalty towards visiting a city such as Cairo through tangible heritage such as the historic mosque, including (1) the expectations or values created before the visit to Cairo, and (2) the visitor's satisfaction with the destination and with the heritage itself.

In this section, the hypotheses of empirical research are established. The main objective of this research is to identify the significant factors that influence the loyalty of tourists who visit destinations with a heritage component, proposing a methodology that analyzes seven hypotheses based on previously analyzed literature:

**Hypothese 1 (H1).** *Expectations of the visit to Cairo have a significant and positive relationship with the expectations of the visit to the Ibn Tulun Historical Mosque.*

**Hypothese 2 (H2).** *Expectations of the visit to the Ibn Tulun Historical Mosque have a significant and positive relationship with satisfaction in Cairo.*

**Hypothese 3 (H3).** *Expectations of the visit to the Historical Mosque Ibn Tulun have a significant and positive relationship with the satisfaction in the visit to the Historical Mosque Ibn Tulun.*

**Hypothese 4 (H4).** *Loyalty to the Ibn Tulun Historical Mosque has a positive and significant relationship with loyalty to Cairo.*

**Hypothese 5 (H5).** *Satisfaction with visiting Cairo has a significant and positive relationship with satisfaction of visiting the Ibn Tulun Historical Mosque.*

**Hypothese 6 (H6).** *Satisfaction with the Ibn Tulun Historical Mosque has a significant and positive relationship with loyalty to the city of Cairo.*

**Hypothese 7 (H7).** *Satisfaction with the Historic Mosque has a significant and positive relationship with loyalty to the Ibn Tulun Historic Mosque.*

The relationships between the different factors that can be observed in the model are shown in Figure 1 (Proposed model).

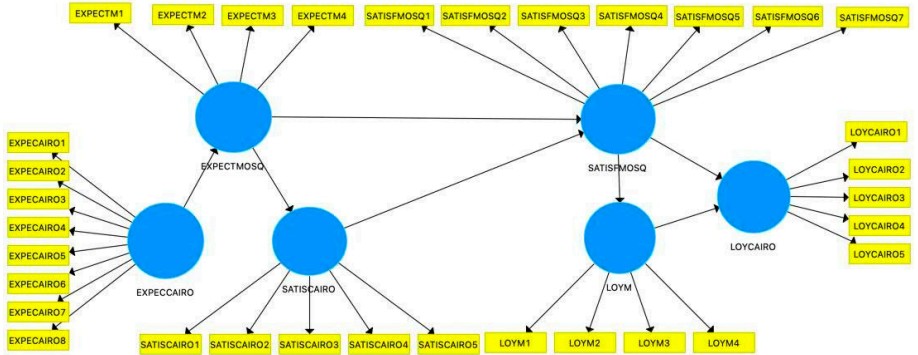

**Figure 1.** Proposed research design. Source: Own elaboration.

## 4. Methodology

### 4.1. Overview of Cairo from the Tourist Point

In Cairo, tourism is considered one of the fastest growing sectors, and has an essential impact on the city's economic development [52]. The age of tourism in Cairo and Egypt in general spans over 150 years. According to global tourism reports, it has been a favorable prominent tourist destination since 1869 [53].

In Egypt, the number of international arrivals was two million in 1989, and this increased to more than 13 million in 2019, with 21% tourist growth [54]. The global reports expected 15 million international arrivals in 2020 before the spread of COVID-19, which affected tourism demand in Cairo [54].

Cultural heritage tourism is a major sector that the Egyptian capital relies on, resulting in numerous ongoing mega tourist heritage projects like the Grand Egyptian Museum (expected to be officially opened by the end of 2022 with huge restoration campaigns on its sites of heritage). Tourism is one of the main contributors to the Egyptian economy. For example, in 2018, the country received 12.7 billion in USD [55].

Although the Egyptian governmental plan in the cultural tourism industry focused more on recreational tourism to avoid repetitive terrorist attacks, Cairo maintained acceptable averages of cultural tourist destinations, due to the high demand on its treasures of heritage—the pyramids, ancient mosques, and churches [56]. Reviving the ability to re-attract significant tourists is a priority for the government, which started to rely on unconventional destinations in Cairo. Many of these unconventional destinations belong to Islamic and Coptic civilizations. These destinations may achieve higher demand to visit the city from tourists, a potential solution to restoring the number of arrivals.

Islamic monuments and landmarks in Cairo encompass a wide range of buildings that bear different styles of art and architecture and are enriched by the influence of the design and construction of structures belonging to Islamic culture [57]. The Islamic architectural types in Cairo include Mosques, Mausoleums, Palaces, gates, houses and citadels, among others. Each type enjoys a variety of styles and architectural elements [58].

Several Islamic monuments and architecture structures in Cairo belong to different periods of Islamic eras, including the alabaster mosque of Mohamed Ali, the mosque of Amr ibn El Aas, the Complex of Qalwun and the Mosque of Aḥmad ibn Ṭūlūn [5].

### 4.2. Ibn Ṭūlūn Mosque "Architecture and History"

The mosque was constructed in 263AH/876-77 AD, and was completed in 265 AH/April-May 879 AD. It was the third congregational mosque built in Egypt during the Islamic era [59]. It is the longest surviving mosque among other old mosques in the Islamic history of Egypt, and one of the oldest intact mosques in the whole Islamic world [20]. It was founded by Aḥmad ibn Ṭūlūn, who was appointed as a governor of Egypt by the Abbasside caliph in Baghdad in 254 AH/867 AD [60].

The mosque lies in one of the historical districts of Cairo, AL Khalifa District, which is cut by the greatest street of Saliba, according to the ancient design of the city of Samraa in Iraq. Nowadays, it extends between two popular shrines and mosques: Sayda Aysha (near Saldin Square) and the mosque of Sayyida Zaynab (in Port Street). The mosque is neighbored from the northwest by the Mamluk mosque and the Madrasa of Prince Sirghitmish from the 14th century, while to its side lies the famous museum and house of the English officer Gayer-Anderson. The main entrance to the mosque points to the northeast side at the street which bears the same name of the founder "Ahmed Ibn Ṭūlūn".

The architecture consists of a large open courtyard centralized by an ablutions fountain and surrounded by four riwaq or porticoes on the sides. The mosque is unique for its huge number of pillars; it has 160 Pillars, 42 doors and 128 windows. The main portico of Mihrab or niche that gives the direction to Mecca has six mihrabs, making it one of only two in the Islamic world to have multiple mihrabs; five of them are stucco flat ones, and two pairs of them are flanked to each other along the axis of the main one, concave style.

The mosque also has a unique, spiral-shaped minaret, which lies in the northern western Zyada, and is connected to the building by a bridge to the roof. Its entrance gives way immediately to the outer stairs. Its spiral shape consists of four distinct stories, and its style resembles that of the great mosque of Samarra in Iraq [20]. The mosque was built of limestone knoll, but mainly red bricks were used, extending up to the ceiling [59].

The mosque is associated with several historical narrations and legends which gave it a special importance. One of these narrations is the non-satisfactory statue, refusing to pray, made by the mosque builders because they did not know the source of money used to construct it [60]. Other stories relate to its positioning on Yashkur hills to be hanging and higher than the ground level. Moreover, the treasure story is widely known as a justification for saving money to build the huge building [61]. There are also stories narrating the Coptic architect of the mosque [61]. Ibn Ṭūlūn wanted to build a mosque that would survive any natural disaster like flooding and earthquakes, and it survives until now as the oldest intact building dated back to the ninth century [4,6].

### 4.3. Tourist Details

The Mosque requires no paid fees to enter, and it is open all day from 8 am until 4 pm, Monday through Sunday, normally even during prayer times. Prayers are practiced in the mosque by the local citizens of Muslims. The average time required to visit it is around two hours.

It is one of the marvelous models of Islamic heritage in Cairo and is considered an example of inspiration and innovation across history. It is not currently included in most of the tourist attractions in the city, so it is missed by thousands of tourists who visit Cairo without passing through the glory of the mosque and its historical value. The mosque suffers from the poor marketing of the tourist operators in travel agencies, which could be interpreted as a problem of "heritage congestion" in Cairo. Therefore, the tourists do not have enough time to visit all treasures in the Egyptian Capital [62].

### 4.4. Questionnaire and Scales

The collection of information was carried out through a questionnaire provided to each tourist after their experience in Cairo and search for its material heritage. The first phase was developed from January to March 2022. The correct validation of the questionnaire and the formulation of the questions was based on consolidated items from previous research.

Once the items were obtained, a two-step refinement process was followed. First, the items proposed by a researcher were analyzed; second, the final questionnaire was tested and verified by the research manager in Cairo. In accordance with this procedure, the adequate validity of the items that form the theoretical model designed in this research was verified twice.

The questionnaire was organized into four sections. The first section included questions about the demographic profile of the tourist. The following sections asked about their expectations of the visit to the city of Cairo, its historical mosques and the satisfaction that the visitor obtained in both cases to led to loyalty, which influenced their intention to return or recommend a visit to the destination and/or visit Egypt (Cairo). The different variables were measured on a five-point Likert-type scale from 1: totally disagree to 5: totally agree. Questions previously tested from other studies, as shown in Table 1, were adapted and used in this research.

**Table 1.** Used scales.

| Autors | Dimension | Indicator |
|---|---|---|
| [14,63–65] | Expectations (EspectCairo;Espectmosq) | (EXPECCAIRO 1) Notoriety, (EXPECCAIRO 2) Tradition and roots, (EXPECCAIRO 3) Liking, (EXPECCAIRO 4) The atmosphere of the establishments you visited was to your liking, (EXPECCAIRO 5) cultural richness, (EXPECCAIRO 6) hospitality, EXPECCAIRO 7) Conservation, (EXPECCAIRO 8) Cleanliness, (EXPECTMOSQ 1) Notoriety, (EXPECTMOSQ 2) Tradition, (EXPECTMOSQ 3) Liking, (EXPECTMOSQ 4) Conservation |
| [66–69] | Loyalty (LOYCAIRO) | (LOYCAIRO 1) I will recommend to family and friends, (LOYCAIRO 2) I will visit Cairo—Egypt again, (LOYCAIRO 3) I will recommend visiting Cairo—Egypt, (LOYCAIRO 4) My next trip will probably be some of the tourist sites in Cairo |
| [69–71] | LoyM (LOYM) | (LOYM 1) Global image of Cairo is positive, (LOYM 2) It was worth visiting Cairo, (LOYM 3) Egypt is a good place to visit, (LOYM 4) Egypt is recognized/appreciated for its gastronomy. |
| [64,72,73] | Satisfaction (SATISCAIRO) | (SATISCAIRO 1) The visit to Cairo was satisfactory, (SATISCAIRO 2) Satisfied with the quality and price ratio, (SATISCAIRO 3) Satisfaction with the environment, (SATISCAIRO 4) Good place to visit with family |
| [14,64,74] | Satisfaction (SATISFMOSQ_) | (SATISFMOSQ 1) The view to the mosque was satisfactory, (SATISFMOSQ 2) Satisfied with the visit in general, (SATISFMOSQ 3) Its inhabitants are friendly and hospitable, (SATISFMOSQ 4) Good place to visit with family, (SATISFMOSQ 5) It has good and useful tourist information offices. (SATISFMOSQ 6) The experience is worth it, (SATISFMOSQ 7) The guide services at the mosque are satisfactory. |

In the second phase, the questionnaire was administered in English, since in Cairo, English is the most used language after Arabic. Before beginning to carry out the questionnaire, the interviewer asked the tourist for his collaboration and provided him with information on the objectives of the investigation. The tourist completed the questionnaire anonymously with complete autonomy. The questionnaire was formed in four parts of the designed theoretical model (Figure 1): gastronomy, quality, satisfaction, and loyalty. At the end of the questionnaire, items on the sociodemographic profile were included. The total number of items used for our model was 31, after the item purification process through the calculation of Cronbach's alpha coefficient for each dimension. The fieldwork was carried out during the months of April to May 2022, through a simple random sampling of visitors seeking to visit traditional mosques in Cairo, Egypt. A pretest of 20 questionnaires was carried out. In total, the number of valid questionnaires was 184.

The total Cronbach's alpha index of the items with a value of 0.911 was calculated. As anything above 0.7 in Cronbach's alpha is considered acceptable, these items are acceptable [75]. The data from this research were analyzed using the structural equation analysis software Smart-PLS 4.0 [76]. This is used as a tool for the analysis of complex interrelationships between observed and latent variables and has been widely used and validated for scientific research in the tourism sector [65,77,78].

## 5. Results

The main conclusions drawn from the fieldwork are described below. Firstly, we assessed the reliability and validity of the proposed model; then, the hypothesis test was carried out.

### 5.1. Assessment of the Reliability and Validity of the Model

The model proposed for this study is shown in Figure 1. In this model, the interaction of the relationships between the research elements was considered. Following this idea, an evaluation of the model was carried out to analyze the validity and reliability of the dimen-

sions and to identify which of them are reflective or formative. The results obtained show that the observed variables measure the previously proposed theoretical constructions [79].

Later, the validity and reliability of the means of the dimensions were ensured before drawing conclusions about the relationships between them [80]. The formative dimensions of the model (loyalty) were evaluated following some recommendations [81]. Firstly, loyalty has been considered a formative indicator in the proposed model [82]. According to other authors [83,84], when formative dimensions are used, it is necessary to determine whether or not there is multicollinearity between the different indicators that make up the dimension. Following this idea, positive tests of convergent validity have been carried out through a redundancy analysis of these dimensions [85]. These tests continued with the evaluation of collinearity, with the calculation of the variance inflation factor (VIF) <5. A value of 5 or less would indicate the non-existence of a high multicollinearity [86]. The indicators of the formative dimensions were evaluated by means of weighting, revealing that they are significant [81], and they are shown in Table 2.

**Table 2.** Individual item reliability (formative).

| | EXTERNAL WEIGHTS | VIF |
|---|---|---|
| **EXPECAIRO1** | 0.146 | 3.669 |
| **EXPECAIRO2** | 0.177 | **2.195** |
| **EXPECAIRO3** | 0.130 | **1.384** |
| **EXPECAIRO4** | 0.092 | **2.298** |
| **EXPECAIRO5** | 0.187 | **2.055** |
| **EXPECAIRO6** | 0.186 | **2.146** |
| **EXPECAIRO7** | 0.172 | **2.726** |
| **EXPECAIRO8** | 0.233 | **2.250** |
| **EXPECTM1** | 0.347 | **2.951** |
| **EXPECTM2** | 0.270 | **1.897** |
| **EXPECTM3** | 0.308 | **2.333** |
| **EXPECTM4** | 0.269 | **1.569** |
| **LOYCAIRO1** | 0.274 | 4.461 |
| **LOYCAIRO2** | 0.272 | 4.346 |
| **LOYCAIRO3** | 0.264 | 4.347 |
| **LOYCAIRO4** | 0.282 | 3.288 |
| **LOYM1** | 0.302 | **1.959** |
| **LOYM2** | 0.232 | **2.414** |
| **LOYM3** | 0.300 | 4.497 |
| **LOYM4** | 0.293 | 4.888 |
| **SATISCAIRO1** | 0.247 | 4.820 |
| **SATISCAIRO2** | 0.274 | 4.408 |
| **SATISCAIRO3** | 0.224 | 3.491 |
| **SATISCAIRO4** | 0.206 | **2.903** |
| **SATISCAIRO5** | 0.213 | **1.900** |
| **SATISFMOSQ1** | 0.180 | 4.491 |
| **SATISFMOSQ2** | 0.186 | 3.203 |
| **SATISFMOSQ3** | 0.168 | 3.270 |
| **SATISFMOSQ4** | 0.184 | **2.971** |
| **SATISFMOSQ5** | 0.135 | **2.093** |
| **SATISFMOSQ6** | 0.189 | 3.445 |
| **SATISFMOSQ7** | 0.153 | **2.225** |

Source: Own elaboration.

The values show the non-existence of collinearity in the variables that make up the loyalty dimension.

*5.2. Composite Reliability (CR)*

The reliability of the questionnaire makes it possible to check whether the indicators really measure the dimensions of the variables. The results in Table 3 indicated that all the dimensions were reliable, since their composite or joint reliability is >0.7. These values were considered "satisfactory to good" because they were between 0.70 and 0.95 [79,85]. The composite reliability index is similar to Cronbach's alpha, with the difference that the composite reliability is not influenced by the number of items on the scale. The accepted level of composite reliability is 0.7 and the most stringent is 0.8 [79]. In our study, the

composite reliability is observed in Table 3. The strictest level of 0.8 is exceeded in the four indicators; therefore, they validate the internal consistency of the model.

**Table 3.** Compound reliability.

| Dimensions | | Compound Reliability |
|---|---|---|
| **EXPECCAIRO** | Expectations of Cairo | **0.878** |
| **EXPECTMOSQ** | Expectations of Mosque | **0.855** |
| **LOYCAIRO** | Loyalty to Cairo | **0.946** |
| **LOYM** | Loyalty to Mosque | **0.908** |
| **SATISCAIRO** | Satisfaction with Cairo | **0.909** |
| **SATISFMOSQ_** | Satisfaction with Mosque | **0.927** |

Source: Own elaboration.

### 5.3. Convergent Validity

Convergent validity is used to determine the degree to which all the items try to measure the same thing, and all the indicators must be highly correlated. The most common measure used to assess convergent validity in PLS-SEM is the average variance extracted (AVE). Using the same base as that used for the individual indicators, a value or AVE of 50% or more means that, on average, the dimension explains more than half of the variance of its own indicator [79,81,87]. Table 4 shows the AVE values, with a value greater than 0.5 in all dimensions. Furthermore, this table shows that all indicators meet these criteria, since diagonal items must be significantly larger than multiform items in the corresponding rows and columns [12].

**Table 4.** Average variance extracted.

| Factor | Average Variance Extracted (AVE) |
|---|---|
| **EXPECCAIRO** | 0.556 |
| **EXPECTMOSQ** | 0.695 |
| **LOYCAIRO** | 0.838 |
| **LOYM** | 0.784 |
| **SATISCAIRO** | 0.726 |
| **SATISFMOSQ_** | 0.696 |

Source: Own elaboration.

### 5.4. Discriminant Validity

This index shows the extent to which a dimension is different from others. The positive acceptance range, the values present in the diagonal, must be significantly higher than those found in the rows and columns. In this research, the discriminant validity is shown in Table 5. The mentioned requirements are fulfilled by the model.

**Table 5.** Discriminant validity.

| | EXPECCAIRO | EXPECTMOSQ | LOYALTYCAIRO | LOYM | SATISCAIRO |
|---|---|---|---|---|---|
| **EXPECTMOSQ** | **0.763** | | | | |
| **LOYALTYCAIRO** | 0.803 | **0.706** | | | |
| **LOYM** | 0.722 | 0.595 | **0.882** | | |
| **SATISCAIRO** | 0.913 | 0.805 | 0.825 | **0.737** | |
| **SATISMOSQUE** | 0.910 | 0.821 | 0.829 | 0.734 | **1.077** |

Source: Own elaboration.

### 5.5. Hypothesis Contrast

5.5.1. Explained Variance ($R^2$)

Once the sample measurement model was validated, the internal model was evaluated. This evaluation was conducted to verify the proposed hypotheses for the relationships between dimensions. To achieve this, the $R^2$ value was calculated and thus, importance of the relationships could be analyzed [79]. The explained variance of the endogenous

dimension was found by analyzing the values calculated for R², and in this way it was possible to find the predictive power of the model [12].

### 5.5.2. Bootstrapping

Later, a Student's t distribution of 499 degrees of freedom (n − 1, where n represents the number of subsamples) was performed to calculate the significance of the Path Coefficients. In this way, the values with a significance of 0.007, 0.000 and 0.010 were obtained. Table 6 shows the relationship to be studied, the effect that this relationship should really follow, the Path Coefficient, the t value and whether the relationship has been supported. It was verified that the seven proposed hypotheses are accepted.

**Table 6.** Hypothesis contrast.

| H | Name | Effect | Path Coefficient | *t* Value | *p*-Valor | Supported? |
|---|------|--------|------------------|-----------|-----------|------------|
| **H1** | **EXPECCAIRO  EXPECTMOSQ** | + | 0.693 | 10.089 | 0.000 | Yes |
| **H2** | **EXPECTMOSQ  SATISCAIRO** | + | 0.711 | 7.731 | 0.000 | Yes |
| **H3** | **EXPECTMOSQ  SATISFMOSQ_** | + | 0.060 | 2.849 | 0.004 | Yes |
| **H4** | **LOYM  LOYCAIRO** | + | 0.537 | 6.308 | 0.000 | Yes |
| **H5** | **SATISCAIRO  SATISFMOSQ_** | + | 0.945 | 54.965 | 0.000 | Yes |
| **H6** | **SATISFMOSQ_  LOYCAIRO** | + | 0.412 | 4.322 | 0.000 | Yes |
| **H7** | **SATISFMOSQ_  LOYM** | + | 0.686 | 8.303 | 0.000 | Yes |

 = relationship between variables; + = prove positive effect on the relationship. Source: own elaboration.

Therefore, Hypothesis H1 (Expectations when visiting Cairo have a significant and positive relationship with expectations when visiting the Historical Mosque Ibn Tulun), Hypothesis H2 (Expectations when visiting the Historical Mosque Ibn Tulun have a significant and positive relationship with satisfaction in Cairo) and Hypothesis H3 (Expectations of visiting the Ibn Tulun Historical Mosque have a significant and positive relationship with satisfaction after visiting the Ibn Tulun Historical Mosque) have been supported (Figure 2). Regarding loyalty, it is verified that loyalty to the Ibn Tulun Historical Mosque has a positive and significant relationship with loyalty to Cairo (H4). With respect to satisfaction, the three hypotheses are met; that is, satisfaction with the historic mosque confirms repetition and recommendation to Cairo (H6) and loyalty to one's own historical heritage (H7).

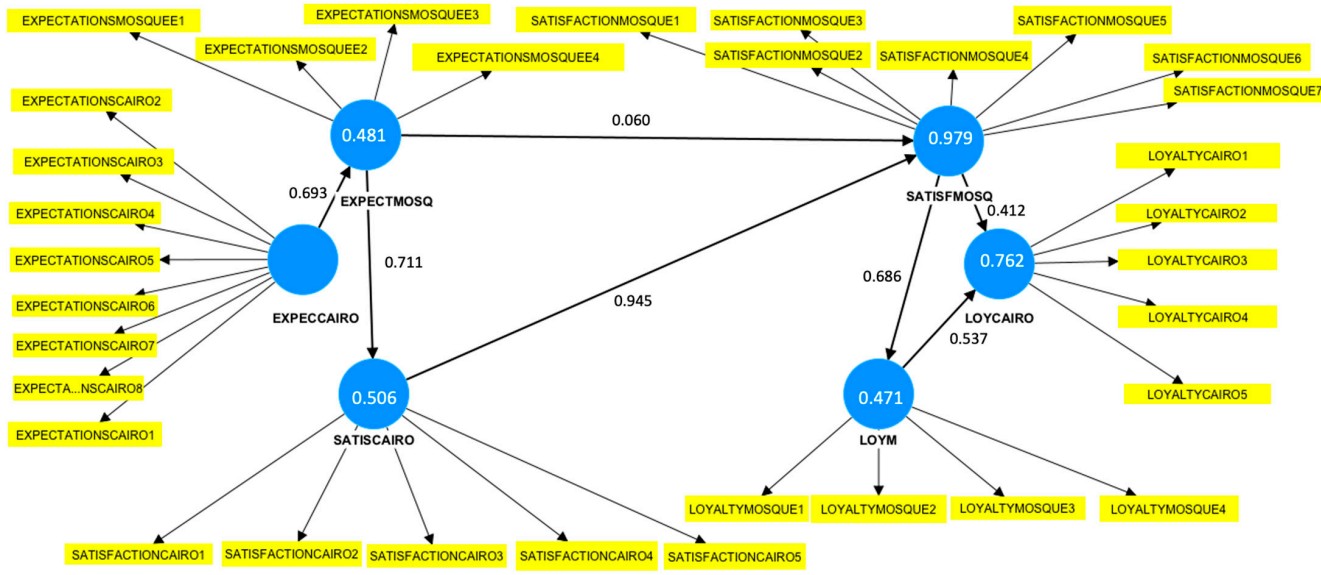

**Figure 2.** Causal relationships model. Source: own elaboration.

## 6. Discussion

This study is based on the growing importance of the study of visitor loyalty, specifically to cities with a strong heritage component and that are recognized for their heritage, but which present other, lesser-known heritage, but no less historical value over time. The most differentiating aspects for the development of tourist loyalty in visits to a World Heritage Site are determined in a city in which different cultures have passed and coexisted, leaving unique and exceptional heritage in the world. The opinions of visitors to the regions during their visit helped to identify the factors that influence loyalty, through satisfaction and expectations both about the city itself and about the historic Ibn Tulun Mosque in order to identify the expectations that are most determinant for the study. In Hypothesis 1 (H1). It was shown that the expectations with the visit to the city influence the expectation about the visit to the Mosque itself. For practical purposes, the manager of the historic mosque must monitor the proper promotion of the property in the context of the city.

As proposed in Hypothesis (H2), it was determined that the expectations of the visit to the Ibn Tulun Historical Mosque have a significant and positive relationship with the satisfaction in Cairo and to satisfaction with the Historical Mosque itself (H3). That is, the final satisfaction of the visitor depends on the expectations. Few authors ratify this fact [88,89], therefore; this study provides a contribution to the literature.

Regarding the variable of loyalty to the Ibn Tulun Historical Mosque, there is a positive and significant relationship with loyalty to Cairo (H4). Therefore, work should be conducted with regard to the recognition of the country's image upon visiting the Mosque. This hypothesis is in line with the report of other authors [90–92].

The variable satisfaction, tested in the hypotheses (H5, H6 and H7), demonstrated that satisfaction with the visit to Cairo has a positive relationship with satisfaction upon visiting the Ibn Tulun Historical Mosque (H5). Satisfaction with the Historical Mosque Ibn Tulun also has a significant and positive relationship with loyalty to the city of Cairo (H6). Lastly, satisfaction with the Historic Mosque has a significant and positive relationship with loyalty to the Ibn Tulun Historic Mosque (H7). These data coincide with the results achieved in other studies on heritage and tourism [93]. Specific heritage and tourist cities [14] verified the positive factors that satisfaction produces, namely the expectations for the intentions of returning to or recommending the destination. For practical purposes, this implies a responsibility for asset managers and destination managers. In the first place, for the continuous improvement of the visit to the historic mosque, the public administrations must ensure continuous satisfaction for visitors to heritage sites, since it will ultimately affect the destination.

Finally, after the end of the global pandemic, the improvement of protection against the coronavirus in heritage tourism management has proven value to visitors. In addition, the loyalty of the destination depends on the measures used to protect visitors against possible contagion and the satisfaction obtained. For this reason, those responsible for heritage sites should include measures to safeguard against the virus in their policies [94] and economic measures to promote tourism [95].

The ACSI model was used in this study to investigate the loyalty of tourists who visit and search for unique heritage, such as the pyramids, but also visit other important material heritage sites that can play an important role in the future of the cultural experience in Cairo. A structural equation was created to study the loyalty of this type of tourists based on their satisfaction with the site.

Tourists visiting Egypt have special characteristics that are important to consider when managing heritage sites, particularly when conducting research on World Heritage sites. This means that the type of heritage related to this typology of tourists' visits must be considered in academic studies and is also important for the organization and management of heritage, as well as future plans for tourism on the site.

### 7. Conclusions and Limitations

Visits to world-renowned cities have advantages and disadvantages in terms of tourism. Among the disadvantages, one of them is hosting lesser-known assets. This study analyzes the so-called historical Mosque of Ibn Tulen. The world heritage cities house sites that can help with visitor loyalty and increase recommendations of the destination. As the results have revealed, factors such as pleasure during the visit, the explanations of the guides and general satisfaction with the visit to Cairo and to the historical Mosque of Ibn Tulen create unique values that are valued by the tourist after the visit. Thus, the visit to the historic mosque of Cairo unites and links the tourist with the destination, positively influencing their loyalty to the territory.

This study has important theoretical and practical implications for heritage management in Egypt. In the first place, the measurement of satisfaction in the visit to Cairo itself, with the ultimate goal of increasing repetition of visits to the country, was researched. In order to foster the loyalty of this type of tourist, it is necessary that the agents involved act on the expectations of the tourists, the level of quality, the satisfaction of the visitor and also their intention to recommend the destination and repeat their visit. Among the conclusions, first, from the most practical perspective, the study suggests how good and useful tourist information offices should be cared for and promoted.

There is a lot of information about the main heritage sites, such as the pyramids in all their extension, and the great museum, tourists are not presented with alternatives and specific information in the tourist promotion brochures, while other much more modern mosques do offer this information. In addition, security for foreign tourists outside attractions should be improved and use of the Spanish language should be considered in tourist areas.

At a theoretical level, carrying out an ACSI model on expectations, satisfaction and loyalty could explain how to include historical tangible heritage in a world-renowned city and how it could contribute to improving tourist loyalty. For city managers and companies, the results corroborated the idea that tangible heritage management could influence future behavior to visit or not visit the country. Tourism policy makers should explore new management methods to provide satisfaction to tourists who seek a memorable experience when experiencing and visiting unique heritage sites. In this sense, the importance of proper information management, preservation, prestige and security should be emphasized.

This investigation, like any research, has potential limitations, the first of which is the sample used. The data were only obtained from tourists who visited Cairo, Egypt, and specifically those who visited the historic Mosque of Ibn Ṭūlūn, which could indicate that the data collected are indicative of a single specific tourist site. This circumstance can be improved in two ways. Firstly, questionnaires could be conducted in other cities in the same country, where there are patrimonies with more than 1000 years that are not so well known. Secondly, a longitudinal study could be conducted, which records the emotions of tourists over time, allowing the measurement between the variables in a more precise way. Among other limitations, the COVID-19 variable was not measured from the perspective of visitor attention and perception. In summary, in terms of future lines of research, the inclusion of other internal and external variables to the subject is recommended, as is a deeper study of the relationship between expectations, satisfaction and destination.

**Author Contributions:** Methodology, R.D.H.-R.; Investigation, I.M.I.E. and L.A.M.Q.; Data curation, R.D.H.-R.; Writing—original draft, I.M.I.E. and R.D.H.-R.; Writing—review & editing, L.A.M.Q. All authors have read and agreed to the published version of the manuscript.

**Funding:** The APC for this manuscript was funded with resolution Nº 048-2023-UNF-DGA by the Institute of Research for the Sustainable Development of Tourism, Universidad Nacional de Frontera, Sullana, Peru.

**Institutional Review Board Statement:** Not applicable.

**Informed Consent Statement:** Informed consent was obtained from all subjects involved in the study.

**Data Availability Statement:** The data presented in this study are available on request from the corresponding author. The data are not publicly available to request permission in the use of authors' data.

**Conflicts of Interest:** The authors declare no conflict of interest.

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
