# Peer review of "Loyalty to World Heritage Cities: A Case Study of the Historic Ibn Ṭūlūn Mosque in Cairo (Egypt)"

_sustainability, doi:10.3390/su151713154_

Round 1
Reviewer 1 Report
Title:
Loyalty to world heritage cities: the case of Cairo (Egypt) and the historic mosque of Ibn Ṭūlūn
First of all, I would like to thank you for the review invitation. I have reviewed this exciting article. This paper states that Cairo city has several unique heritages in the world. As a city, it presents two world inscriptions of humanity. Its uniqueness, pull effect and history is significant, therefore it is essential to deepen its management from a tourism perspective. The main objective of this research is to analyze satisfaction in historic mosques, in the context of heritage tourism. The study focuses on tourist loyalty from various perspectives: expectations, satisfaction with the city itself and the mosque together with loyalty. The fieldwork was carried out in Cairo (Egypt), specifically in the Ibn Tulun Mosque. It is more than 2,000 years old and is considered the oldest mosque in Cairo and one of the most amazing in the world.
Abstract
The abstract quality is not high standard to meet the merit of scientific writing that needs a high standard of writing to publish in outstanding journals like SUSTAINABILITY. Please revise the whole article and remove English grammar problems. I suggest the authors take English editing services from some agencies to improve the quality of this study.
Introduction section
I suggest that authors to read the suggested studies add the latest citations to the introduction, literature and method sections to enhance the quality of the study.
Wang, S., Al-Sulaiti, K., & Shah, S. A. R. (2023). The Impact of Economic Corridor and Tourism on Local Community’s Quality of Life under One Belt One Road Context. Evaluation Review, 47(3), 445-454.
Literature section:
Add literature section. You cannot delete this section. Read the suggested studies and cite these papers in the literature to enhance the quality of your work.
Shah, S. A. R., Zhang, Q., Tang, H., & Al-Sulaiti, K. I. (2023). Waste management, quality of life and natural resources utilization matter for renewable electricity generation: The main and moderate role of environmental policy. Utilities Policy, 82, 101584. https://doi.org/10.1016/j.jup.2023.101584
Materials and Methods
This section is very weak. Please follow the suggested studies and improve your paper. The authors need to improve this section. I am recommending some good studies. Read the methods of these studies, improve your paper, and cite these studies in this section. Suggested useful articles citations:
Abaalzamat, K. H., Al-Sulaiti, K. I., Alzboun, N. M., and Khawaldah, H. A. (2021). "The Role of Katara Cultural Village in Enhancing and Marketing the Image of Qatar: Evidence From TripAdvisor." SAGE Open, 11(2), 21582440211022737.
Local Burden of Disease, H. I. V. C. (2021). Mapping subnational HIV mortality in six Latin American countries with incomplete vital registration systems. BMC Medicine, 19(1), 4. doi:10.1186/s12916-020-01876-4
Discussion section:
I suggest you to discuss the current situation of the COVID-19 pandemic. How it has affected tourism activities. Read the proposed studies to improve your analysis. See the recommended studies and improve your sections.
Khalid, A. S., Al Khulaifi, A., and Al Khatib, F. (2005). "Banking services and customer's Satisfaction in Qatar: A statistical analysis." Studies in Business and Economics, 11(1).
Implications
Explain this section effectively. It needs a better presentation related to the study topic.
Limitations
Discuss the study’s limitations with a separate heading and discuss it briefly.
Policy recommendations
Policy recommendations are not sufficient at this stage of the manuscript. The authors must add a separate section for policy recommendations in the conclusion section. Also, add some exciting limitations regarding political factors for future studies.
Conclusion
The conclusion section needs improvement and the authors need to expand it as it will improve the quality of this study. The English level needs some improvement to reach a satisfactory level, specifically the grammar. It should sufficiently meet quality to reach scientific merit for publication. I recommend that the authors describe the study's scientific contribution to the existing body of knowledge in the discussion section. How does this study’s implications provide useful information for the scientific readership? I endorse this manuscript for publication after minor corrections, as suggested.
Moderate editing of English is the language required.
Author Response
Dear editor and reviewer,
Thank you very much for giving us the opportunity to revise and resubmit the article to sustainability. We appreciate this opportunity since it has allowed us to improve the article by enhancing its consistency, clarifying the empirical analysis, and improving the conclusions, among other aspects. To address the reviewer´s concerns, we have made changes to the Abstract, introduction section, literature section, Methodology, Results, and Discussion sections, implications. Furthermore, following the reviewer´s suggestion, we have reflected on the relevance of the contribution. A more detailed response to the reviewer´s comments is shown below.
RESPONSE TO REVIEWER 1
R1 The abstract quality is not high standard to meet the merit of scientific writing that needs a high standard of writing to publish in outstanding journals like SUSTAINABILITY. Please revise the whole article and remove English grammar problems. I suggest the authors take English editing services from some agencies to improve the quality of this study.
We highly appreciate all your comments and suggestions since they have significantly contributed to the improvement of the research. Further changes have been made to address your concerns in this round of revision. We modify the English of the abstract
R1 Introduction section. I suggest that authors to read the suggested studies add the latest citations to the introduction, literature and method sections to enhance the quality of the study.
Wang, S., Al-Sulaiti, K., & Shah, S. A. R. (2023). The Impact of Economic Corridor and Tourism on Local Community’s Quality of Life under One Belt One Road Context. Evaluation Review, 47(3), 445-454.
We appreciate your comment, which has led us to realize the need to better justify introduction. We have introduced the paragraph in the introduction:
“Likewise, the increase in pollution in cities is due to the lack of adequate mitigation options in the tourism sector. This aspect must be taken into account in cities wishing to increase and improve the quality of tourism, which is a crucial challenge [11]”.
R1 Literature section:
Add literature section. You cannot delete this section. Read the suggested studies and cite these papers in the literature to enhance the quality of your work.
Shah, S. A. R., Zhang, Q., Tang, H., & Al-Sulaiti, K. I. (2023). Waste management, quality of life and natural resources utilization matter for renewable electricity generation: The main and moderate role of environmental policy. Utilities Policy, 82, 101584. https://doi.org/10.1016/j.jup.2023.101584
Thank you very much for your contribution, the authors have introduced this author to improve the section.
“Finally, concepts inherent to heritage and tourism such as sustainability, renewable electricity generation, and environmental concerns must be taken into account [22]”.
R1 Materials and Methods
This section is very weak. Please follow the suggested studies and improve your paper. The authors need to improve this section. I am recommending some good studies. Read the methods of these studies, improve your paper, and cite these studies in this section. Suggested useful articles citations:
Abaalzamat, K. H., Al-Sulaiti, K. I., Alzboun, N. M., and Khawaldah, H. A. (2021). "The Role of Katara Cultural Village in Enhancing and Marketing the Image of Qatar: Evidence From TripAdvisor." SAGE Open, 11(2), 21582440211022737.
Local Burden of Disease, H. I. V. C. (2021). Mapping subnational HIV mortality in six Latin American countries with incomplete vital registration systems. BMC Medicine, 19(1), 4. doi:10.1186/s12916-020-01876-
Thank you very much the authors we have added this autor (Abaalzamat et al 2021), Line 306
R1 Discussion section:
I suggest you to discuss the current situation of the COVID-19 pandemic. How it has affected tourism activities. Read the proposed studies to improve your analysis. See the recommended studies and improve your sections.
Al Sulaiti, K.; Al Khulaifi, A.; Al Khatib, F. Banking Services and Customer’s Satisfaction in Qatar: A Statistical Analysis. Studies in Business and Economics 2005, 11.
Thank you very much, we have added the following paragraph using also two more authors to the text.
“Finally, after the end of the global pandemic, the improvement and care against the coronavirus in heritage tourism management is a proven value for visitors. In addition, the image of the destination depends on the measures used against possible contagion and the satisfaction obtained. For this reason, those responsible for heritage sites should include measures to safeguard against the virus in their policies [94] and economic measures according to the contribution to tourism [95]”.
R1 Implications
Explain this section effectively. It needs a better presentation related to the study topic.
R1 Limitations
Discuss the study’s limitations with a separate heading and discuss it briefly.
Thanks for the appreciation, the authors have separated and limited the "discussion" section
R1 Policy recommendations
Policy recommendations are not sufficient at this stage of the manuscript. The authors must add a separate section for policy recommendations in the conclusion section. Also, add some exciting limitations regarding political factors for future studies.
Conclusion
The conclusion section needs improvement and the authors need to expand it as it will improve the quality of this study. The English level needs some improvement to reach a satisfactory level, specifically the grammar. It should sufficiently meet quality to reach scientific merit for publication. I recommend that the authors describe the study's scientific contribution to the existing body of knowledge in the discussion section. How does this study’s implications provide useful information for the scientific readership? I endorse this manuscript for publication after minor corrections, as suggested.

Reviewer 2 Report
Tourism plays a very important role in the economy of any country in the world, because it contributes to the development of the country's economy and provides the population with jobs. Egypt is a very popular tourist country thanks to its rich cultural heritage. Loyalty to world heritage sites among tourists in Cairo and the Historic Mosque of Ibn Ṭūlūn is the subject of this article.
The main goal of this study is to analyze satisfaction with historical mosques in the context of cultural tourism. The authors provided a thorough analysis of previous studies, described the city of Cairo (Egypt) and the Historic Mosque of Ibn Ṭūlūn, developed research hypotheses. The research is based on a survey of visitors to Cairo (Egypt) and the Historic Mosque of Ibn Ṭūlūn about the loyalty of tourists from different points of view: expectations, satisfaction with the city and the mosque itself, along with loyalty. Processing of the results of the survey of mosque visitors was carried out using the structural equation model of partial least squares (PLS). The obtained results confirm that the satisfaction of visiting the Ibn Tulun Mosque has a positive effect on the loyalty of the tourist, so it is recommended to return and visit.
The article is written at a fairly high scientific and methodical level, but the drawing of the Historic Mosque of Ibn Ṭūlūn is missing for better clarity.
.
Author Response
Dear editor and reviewer,
Thank you very much for giving us the opportunity to revise and resubmit the article to sustainability.
RESPONSE TO REVIEWER 2
We highly appreciate all your comments and suggestions since they have significantly contributed to the improvement of the research. Your comments especially encourage us to continue research in favor of heritage and tourism from the point of view of management.

Reviewer 3 Report
An important sentence in the paper is “the high expectations for tourist satisfaction this makes it harder for these types of cities to stand out, since the higher the expectations, the more difficult it is to surprise those who visit” and I agree with it. Cairo is a true example and proof of this claim. The authors made clear hypotheses, which arose from the excellent topic of the paper. Figures depict the complex relationships that are the theme of the paper. The questionnaires are clearly defined and adequately interpreted, and the limitations of this research are also stated. Very nice and interesting paper.
Author Response
Dear editor and reviewer,
Thank you very much for giving us the opportunity to revise and resubmit the article to sustainability. We appreciate this opportunity since it has allowed us to improve the article by enhancing its consistency, clarifying the empirical analysis, and improving the conclusions, among other aspects.
RESPONSE TO REVIEWER 3
We appreciate your comment. It encourages us to continue conducting tourism and heritage research. Thanks a lot

Round 2
Reviewer 1 Report
I have reviewed the revised version of this manuscript. I am satisfied with the revised manuscript in it's current format. I endorse this paper for publication in it's current format without further changes. Good luck
It is acceptable. However, the authors can enhance its quality.